# SIRT1 Activation Enhancing 8,3′-Neolignans from the Twigs of *Corylopsis coreana* Uyeki

**DOI:** 10.3390/plants10081684

**Published:** 2021-08-16

**Authors:** Hyun-Woo Kim, Jin-Bum Jeon, Mi Zhang, Hyo-Moon Cho, Byeol Ryu, Ba-Wool Lee, William H. Gerwick, Won-Keun Oh

**Affiliations:** 1Korea Bioactive Natural Material Bank, Research Institute of Pharmaceutical Sciences, College of Pharmacy and Research Institute of Pharmaceutical Sciences, Seoul National University, Seoul 08826, Korea; hwkim8906@gmail.com (H.-W.K.); blyu03@snu.ac.kr (J.-B.J.); mintazhang@snu.ac.kr (M.Z.); chgyans@naver.com (H.-M.C.); estrella56@snu.ac.kr (B.R.); paul36@snu.ac.kr (B.-W.L.); 2Center for Marine Biotechnology and Biomedicine, Scripps Institution of Oceanography, University of California, La Jolla, CA 92093, USA; wgerwick@health.ucsd.edu; 3Skaggs School of Pharmacy and Pharmaceutical Sciences, University of California San Diego, La Jolla, CA 92093, USA

**Keywords:** *Corylopsis coreana*, hamamelidacease, neolignan, SIRT1

## Abstract

Three undescribed 8,3′-neolignans, corynol (**1**), 3-methoxy-corynol (**2**) and 3′-deoxy-corynol (**3**), together with two bergenin derivatives, three flavonoids, two hydrolysable tannins and six simple phenolic compounds, were isolated from the twigs of *Corylopsis coreana* Uyeki. The structures of the 8,3′-neolignans were elucidated by analyzing their NMR, HRESIMS and ECD spectra. All the isolated compounds were evaluated for their SIRT1 stimulatory activity, and 3′-deoxy-corynol (**3**) showed SIRT1 stimulation activity. Furthermore, a docking study of **3** was performed with three representative binding pockets of SIRT1.

## 1. Introduction

The genus *Corylopsis* is one of 31 genera of the Hamamelidacea family, and it is distributed primarily in tropical and subtropical mountainous regions. *Corylopsis coreana* Uyeki, also called Korean winter hazel, is an endemic species of the Korean Peninsula that is distributed in the Jiri Mountains and Gwangdeok of South Korea. It is a deciduous shrub with yellowish brown or dark brown twigs that is famous for flowering before the leaves appear in the spring. Little research has been performed on *C. coreana*; however, several bioactivities have been reported for extracts of the plant, including antioxidative, antiproliferative, antimicrobial and xanthine oxidase inhibitory activities [1,2,3]. Hydrolysable tannins and flavonoids have been reported as chemical constituents of *C. coreana*, and this plant produces large quantities of bergenin, the C-glycoside of 4-*O-*methyl gallic acid [4,5].

Silent mating type information regulation 2 homolog 1 (SIRT1), a member of the sirtuins, is a nicotinamide adenine dinucleotide (NAD)-dependent deacetylase that is involved in cellular metabolism [6]. Over the past two decades, accumulating evidence has indicated that sirtuins are not only energy status sensors but also protect cells against metabolic stresses such as regulating inflammation, aging and autophagy [7]. SIRT1 has a key role in metabolic health by deacetylating many target proteins in numerous tissues, including liver, muscle, adipocyte and heart tissue [8]. Recently, SIRT1 research has focused on regulating cellular senescence and delaying cellular aging by deacetylation of the p53 protein [9].

Despite the biological interests in SIRT1, only a few natural product-derived SIRT1 activators have been reported. Resveratrol, a well-known stilbenoid antioxidant, is a SIRT1 activator that stabilizes protein–substrate interactions [10]. In addition, several natural products, including quercetin and fisetin (flavonoids), berberine (isoquinoline alkaloid) and curcumin (diarylheptanoid), have been shown to directly or indirectly increase SIRT activation in cells [11]. In our previous studies, we reported that flavonoids from *Psoralea corylifolia* possess SIRT1-promoting activities [12]. Interestingly, most of these reported natural SIRT1 activators have a diphenyl moiety in their structure.

Lignans are a major class of polyphenol and commonly possess a diphenyl moiety as part of their structure; however, they have not been studied for their SIRT1 stimulatory effects. The only exception is magnolol, which has been reported to increase SIRT1 expression in an experimental rat stroke model [13]. Lignans and neolignans are a large group of naturally occurring phenols that are widely distributed within plants [14], and some 5258 lignans have been reported in nature. Among them, only 83 compounds have been reported as 8,3′-neolignans. Biosynthetically, 8,3′-neolignans are reduced forms of 7,4′-epoxy-8,3′-neolignans, a process that is catalyzed by phenylcoumaran benzylic ether reductase (PCBER) [15], and also found in the metabolism of dehydrodiisoeugenol (DDIE) with demethylation and ring-opening reaction [16]. The bioactivity of 8,3′-neolignans has rarely been investigated. Icariside E5, which was isolated from *Capsicum annuum*, protects Jurkat cells from apoptosis induced by oxidative stress mediated by serum withdrawal [17]. Piperkadsin C, which was isolated from *Piper kadsura*, potently inhibits NO production in LPS-activated BV-2 microglial cells. Thus, investigations of the bioactivities of 8,3′-neolignan could be worthwhile and may lead to the discovery of new sources of bioactive compounds.

Investigations into the natural product chemical diversity in *C. coreana* isolated three previously undescribed 8,3′-neolignans (**1**–**3**) with 13 known compounds (**4**–**16**). The structures of the undescribed compounds were identified and named corynol (**1**), 3-methoxy-corynol (**2**) and 3′-deoxy-corynol (**3**) based on an analysis of their NMR, HRESIMS and ECD spectra. Subsequently, we evaluated the isolated compounds for their SIRT1-promoting activities.

## 2. Results and Discussion

### 2.1. General

The methanolic extract of twigs of *C. coreana* was suspended in H_2_O and partitioned successively using *n*-hexane, CH_2_Cl_2_, EtOAc and H_2_O. The water residues were subjected to Diaion HP-20 resin column chromatography and eluted with H_2_O and MeOH to provide two fractions. The CHCl_2_, EtOAc and MeOH extracts were separated by chromatographic methods, and three previously undescribed neolignans (**1**–**3**) were isolated along with 13 known compounds including bergenin (**4**) [18], 11-*O*-galloyl bergenin (**5**) [19], (−)-epicatechin (**6**) [20], (−)-epicatechin 3-*O*-gallate (*7*) [21], (−)-epigallocatechin gallate (**8**) [22], 1,2,4,6-tetra-*O*-galloyl-β-D-glucose (**9**) [23], 1,2,3,4,6-penta-*O*-galloyl-β-D-glucose (**10**) [24], ellagic acid (**11**) [25], benzoic acid (**12**) [26], 3,4,5-trihydroxybenzoic acid (**13**) [27], 3,5-dihydroxy-4-methoxybenzoic acid (**14**) [28], 1-(4-hydroxy-3,5-dimethoxyphenyl)-1-propanone (**15**) [29] and 3,4,5-trimethoxyphenyl-(6′-*O*-galloyl)-*O*-β-D-glucopyranoside (**16**) (Figure 1) [30]. The structures of the known compounds (**4**-**16**) were identified by comparison with previously reported NMR data.

### 2.2. Structure Elucidation of the Previously Undescribed Compounds

Compound **1** was obtained as a pale brown amorphous powder. Its molecular formula was determined to be C_18_H_20_O_3_ based on the high-resolution electrospray ionization mass spectrometry (HRESIMS) ion peak at *m*/*z* 283.1332 [M-H]^−^ (calcd for C_18_H_19_O_3_, 283.1334). The IR spectrum showed the presence of hydroxy (3365 cm^−1^) and aromatic (1441 cm^−1^) groups. In the ^1^H NMR spectrum of **1** (Appendix A), two aromatic proton signals at *δ*_H_ 6.95 (2H, d, *J* = 8.4 Hz, H-2, 6) and 6.64 (2H, d, *J* = 8.4 Hz, H-3, 5) showed an AA′XX′ aromatic system (A ring). The meta-coupled proton signals at *δ*_H_ 6.47 (1H, d, *J* = 1.9 Hz, H-6′) and 6.44 (1H, d, *J* = 1.9 Hz, H-2′) indicated the presence of a meta-coupled aromatic system (B ring) in **1**. Three olefinic proton signals at *δ*_H_ 5.90 (1H, ddt, *J* = 16.8, 10.2, 6.7 Hz, H-8′), *δ*_H_ 5.00 (1H, dd, *J* = 16.8, 1.1 Hz, H-9′a) and *δ*_H_ 4.97 (1H, dd, *J* = 10.2, 1.1 Hz, H-9′b) indicated the presence of an unsaturated aliphatic chain.

Further analysis of the 2D NMR spectra of **1** helped to define its molecular structure (Figure 2 and Appendix A). The connection of the saturated aliphatic chain between H-7 (*δ*_H_ 2.88 and 2.55), H-8 (*δ*_H_ 3.33) and H-9 (*δ*_H_ 1.11) was confirmed as a propyl group by analysis of the ^1^H-^1^H COSY spectrum. The connection of an unsaturated aliphatic chain between H-7′ (*δ*_H_ 3.19), H-8′ (*δ*_H_ 5.90) and H-9′ (*δ*_H_ 5.00 and 4.97) was also confirmed as a propylene group by COSY. The HMBC correlations of H-8 (*δ*_H_ 3.33) with C-1 (*δ*_C_ 133.7), C-2′ (*δ*_C_ 119.0) and C-4′ (*δ*_C_ 141.9) suggested that the aromatic rings were linked via a saturated aliphatic chain at C-1 and C-3′. The propylene group was located at C-1′, which was confirmed by the HMBC correlation of H-7′ (*δ*_H_ 3.19) with C-6′ (*δ*_C_ 113.7) and C-2′ (*δ*_C_ 119.0). From these results, the structure of compound **1** was assigned as an 8,3′-neolignan.

The absolute configuration of **1** was established by analysis of its ECD spectrum. 1-Deoxycarinatone, which has the same 8,3′-neolignan structure as that of **1**, showed a negative Cotton effect at 285 nm in its ECD spectrum [31,32]. Since the ECD spectrum of **1** also showed negative Cotton effects at 220 and 285 nm (Appendix A), the absolute configuration of **1** was determined to be 8*S*. Thus, compound **1** was characterized as 8(*S*)-4,4′,5′-trihydroxy-8,3′-neolign-8′-ene (corynol); this represents the first description of this compound from nature.

Compound **2** was isolated as pale brown amorphous powder. By negative ion mode HRESIMS, it showed a protonated molecular ion peak at *m*/*z* 313.1485 [M-H]^−^ (calcd for C_19_H_21_O_4_, 313.1440), and thus the molecular formula of **2** was determined to be C_19_H_22_O_4_. The IR spectrum showed the presence of hydroxy (3399 cm^−1^), aromatic (1603, 1514 and 1446 cm^−1^) and methoxy (1367 cm^−1^) groups. The ^1^H NMR spectrum of **2** was similar to that of **1**, but the ABX-patterned aromatic proton signals at *δ*_H_ 6.63 (1H, dd, *J* = 8.0, 1.2 Hz, H-5), 6.61 (1H, brs, H-2) and 6.57 (1H, d, *J* = 8.0 Hz, H-6) and one methoxy proton signal at *δ*_H_ 3.73 (3H, s, 3-OCH_3_) were different from those of **1** (Appendix A). The location of the methoxy group was confirmed as C-3 on the A ring as an HMBC correlation was observed between the methoxy protons (*δ*_H_ 3.73) and C-3 (*δ*_C_ 148.4) (Figure 2 and Appendix A). The HMBC correlations of H-8 (*δ*_H_ 3.35) with C-1 (*δ*_C_ 134.3), C-4′ (*δ*_C_ 142.0) and C-2′ (*δ*_C_ 119.2) suggested that compound **2** was also an 8–3′-linked neolignan (Figure 2). From these results, compound **2** was determined as a 3-methoxylated derivative of corynol (**1**). The absolute configuration of **2** was confirmed as 8*S* from a negative Cotton effect at 220 and 285 nm in its ECD spectrum (Appendix A). Thus, the structure of compound **2** was identified as 8(*S*)-3-methoxy-4,4′,5′-trihydroxy-8,3′-neolign-8′-ene (3-methoxycorynol, **2**).

Compound **3** was obtained as a pale brown amorphous powder, and the molecular formula was determined to be C_18_H_20_O_2_ from the HRESIMS data. The presence of hydroxy (3355 cm^−1^) and aromatic (1613, 1509, and 1436 cm^−1^) groups was suggested from the IR spectrum of **3**. The ^1^H NMR spectrum of **3** was similar to that of **1**, including an aromatic region in which AA′XX′ patterned proton signals at *δ*_H_ 6.93 (2H, d, *J* = 7.9 Hz, H-2,6) and *δ*_H_ 6.63 (2H, d, *J* = 7.9 Hz, H-3,5) were observed, but the ABX patterned proton signals at *δ*_H_ 6.88 (1H, d, *J* = 2.0 Hz, H-6′), 6.78 (1H, d, *J* = 8.3, 2.0 Hz, H-2′) and 6.65 (1H, d, *J* = 8.3 Hz, H-5′) were different (Appendix A). These results suggested that one hydroxy group was absent from the B ring of **1**. The structure of compound **3** was confirmed as an 8–3′-linked neolignan by analysis of the HMBC spectrum (Figure 2 and Appendix A), which showed correlations of H-8 (*δ*_H_ 3.33) with C-1 (*δ*_C_ 133.6), C-4′ (*δ*_C_ 153.9) and C-2′ (*δ*_C_ 127.5). The absolute configuration of **3** was confirmed as 8*S* by comparing its ECD spectrum to that of **1** (Appendix A). From these results, the structure of compound **3** was characterized as 8(*S*)-4,4′-dihydroxy-8,3′-neolign-8′-ene (3′-deoxycorynol).

### 2.3. Evaluation of SIRT1 Stimulatory Effects

All isolated compounds were evaluated for their SIRT1 stimulatory effects by measuring the deacetylation level of p53 using a SIRT1-p53 luciferase assay. For this cell-based assay, HEK293 cells were transfected with reporter plasmids, PG13-luc (wt p53 binding sites), along with the plasmid encoding myc-tagged p53 (myc-p53) and the plasmid encoding flag-tagged SIRT1 (flag SIRT1) with RSV-β-gal plasmid as an internal control. If the test molecules enhanced SIRT1 activity, p53 binding to reporter plasmids would be decreased by SIRT1 regulation, and the expression of luciferase from the reporter plasmid would also be decreased. Among the isolated compounds, **3**, **10** and **16** showed SIRT1 stimulating activities (Figure 3A) and no cytotoxicity at a concentration of 10 μM (Appendix A). However, it has been reported that gallotannins such as compounds **10** and **16** attenuate SIRT1 expression in cells [33]. Accordingly, we focused on compound **3** and its SIRT1 stimulatory effects (Figure 3B). To confirm the direct effect of compound **3** on SIRT1 deacetylation activity, the actual level of substrate used in SIRT1 activity was measured using an NAD^+^/NADH assay. The NAD^+^/NADH ratio from whole-cell extracts of SIRT1 overexpressed HEK293 cells increased after treatment with compound **3** in a dose-dependent manner (Figure 3C). Accordingly, treatment with compound **3** enhanced SIRT1 deacetylation activity and attenuated p53 transcriptional activity.

To investigate the mechanism of the compound **3**-induced SIRT1 stimulatory effect, its direct enzymatic effect on SIRT1 was measured with the p53 peptide substrate in a bioluminescence assay. As shown in Figure 3D, the SIRT1 reactions showed a one-and-a-half-fold increase in the deacetylated peptide substrate after treatment with compound **3** (20 μM). This result indicates a direct enhancement of SIRT1 activity by compound **3**.

To investigate the direct stimulatory mechanism of compound **3**, molecular docking studies between SIRT1 and compound **3** were carried out with three different crystal structures of the human SIRT1 protein (4KXQ, 4I5I and 5BTR). 4KXQ represents the heterodimeric (chains A and B), closed conformation of SIRT1 which is bound to adenosine-5-diphosphoribose (APR), whereas 4I5I represents the dimeric (chains A and B) conformation of SIRT1 bound to NAD or the carboxamide SIRT1 inhibitor. 5BTR represents the heterotrimeric (chains A–C for SIRT1 and D–F for p53), closed conformation of SIRT1 bound to resveratrol. The docking results were manipulated using the GBVI/WSA dG scoring function with the generalized Born solvation model (GBVI). The GBVI/WSA dG function is a force field-based scoring system that estimates the free energy of binding of the ligand from a given orientation. The interaction results were evaluated with the S score. A lower S score indicates a stronger interaction with the binding site. Compound **3** showed S scores of −6.5323 for 4KXQ, −6.4853 for 4I5I, and −5.7650 for 5BTR. In the 4KXQ model, the two aromatic ring systems of **3** interact with the Arg466 and Ser441 binding sites via CH-π interactions (Figure 4A). In the 4I5I binding site, Phe273 and Phe297 interact with the A ring and H-8 of **3** via π-π and CH-π interactions (Figure 4B). In the 5BTR structure, hydrogen bonding interactions were observed between Glu416 and 4-OH, and CH-π interactions were observed between Phe414 and the A ring of **3** in the resveratrol binding pocket (Figure 4C). These results suggested that compound **3** binds with a low S score to the SIRT1 binding pocket/APR binding site of 4KXQ.

## 3. Materials and Methods

### 3.1. General Experimental Procedures

Optical rotations were measured on a JASCO P-2000 polarimeter (JASCO, Easton, MD, USA). All UV and ECD spectra were recorded with a Chirascan ECD spectrometer (Applied Photophysics, Leatherhead, UK). IR spectra were recorded on a JASCO FT/IR-4200 spectrometer. NMR spectra were acquired on a JEOL GSX 400 (JEOL, Tokyo, Japan) and AVANCE-600 NMR spectrometer (Bruker, Billerica, MA, USA). All HRESIMS data were measured on a Waters Xevo G2 QTOF mass spectrometer (Waters Co., Milford, MA, USA). Column chromatography was performed with Diaion HP-20 (Mitsubishi Chemical Industries Ltd., Tokyo, Japan) and Sephadex LH-20 (25–100 μm, GE Health care, IL, USA). Thin layer chromatography (TLC) was carried out using Kieselgel 50 F254 coated normal-phase silica gel TLC plates (Merck, Darmstadt, Germany). The preparative HPLC system was equipped with a G-321 pump (Gilson, Middleton, WI, USA), a G-151 UV detector (Gilson) and a YMC-Triart C_18_ column (250 mm × 10 mm I.D., 5 μm, YMC Co., Ltd., Koyoto, Japan). All solvents were purchased from Daejung Chemicals & Metals CO. Ltd. (Siheung, Korea).

### 3.2. Plant Material

Twigs of *C. coreana* (2.0 kg) were collected from the Herbarium of the Medicinal Plant Garden, Seoul National University, Goyang, Korea, in July 2015 and authenticated by Prof. Dr. Tae-Jin Yang (College of Agricultural and Life Sciences, Seoul National University). The voucher specimen (SNU-1507) of the plant was deposited in the Herbarium of the Medicinal Plant garden of the College of Pharmacy, Seoul National University.

### 3.3. Extraction and Isolation

Air-dried C. coreana twigs (2.0 kg) were extracted with 8 L of methanol, three times, 90 min each by ultrasonication. The total extract (78 g), which was evaporated in vacuo, was diluted with distilled water (3 L), followed by fractionation with the *n*-hexane fraction (2 g), methylene chloride fraction (3.2 g), ethyl acetate fraction (11 g) and aqueous residues. The aqueous residues were subjected to HP-20 resin column chromatography eluting with water (49 g) and methanol (12.2 g) to provide two fractions.

The methylene chloride fraction (3.2 g) was subjected to silica gel column chromatography eluted with mixtures of CH_2_Cl_2_-MeOH (100:0→50:50→0:100, *v*/*v*) to give 20 fractions (S1–S20). Fraction S13 was separated on a semi-preparative HPLC column with a MeCN-H_2_O isocratic system (50:50, *v*/*v*), yielding compounds **1** to **3**.

The ethyl acetate fraction (11.0 g) was subjected to reversed phase C_18_ MPLC at a flow rate of 80 mL/min with a step gradient of MeCN-H_2_O with 0.1% formic acid (10:90–80:20, *v*/*v*) to obtain eight fractions (EA1–EA8). Compound **11** was crystallized from the water residue subfractions EA1-3. Fraction EA2 was applied to semi-preparative HPLC at a flow rate of 4 mL/min with MeCN-H_2_O using a 0.1% formic acid isocratic system (40:60, *v*/*v*) to obtain compounds **4**, **6** and **14**. Fraction EA4 was further purified by semi-preparative HPLC at a flow rate of 4 mL/min with MeCN-H_2_O using 0.1% formic acid (35:65, *v*/*v*) to afford compounds **7**, **8** and **9**. Subfraction EA5 was applied to Sephadex LH-20 column chromatography with MeOH to yield 11 subfractions (EA5-1–EA5-11). These subfractions were characterized by thin layer chromatography. Semi-preparative HPLC was performed to isolate compound **13** from EA5-1 at a flow rate of 4 mL/min using MeOH-H_2_O with a 0.1% formic acid isocratic system (30:70, *v*/*v*). Compound **10** was isolated from EA5-3 by semi preparative HPLC with a MeCN-H_2_O isocratic system (40:60, *v*/*v*) at a flow rate of 4 mL/min.

The methanol fraction (12.2 g) from HP-20 resin chromatography was subjected to reversed phase C18 MPLC at a flow rate of 60 mL/min eluting with step gradient systems of MeCN-H_2_O with 0.1% formic acid (10:90–50:50, *v*/*v*) to yield 11 fractions (M1–M10). Subfraction M2 was purified by Sephadex LH-20 column chromatography using MeOH to obtain 12 subfractions (M21–M212). These subfractions were characterized by thin layer chromatography. Compound **16** was acquired by preparative HPLC at a flow rate of 4 mL/min using a MeOH-H_2_O isocratic system (40:60, *v*/*v*) along with Sephadex LH-20 column chromatography with MeOH. The M28 subfraction was purified by semi-preparative HPLC with MeCN-H_2_O with a 0.1% formic acid isocratic system (30:70, *v*/*v*) and subjected to additional semi-preparative purification with MeOH-H_2_O with a 0.1% formic acid isocratic system (17:83, *v*/*v*) to yield compound **5**. Subfraction M5 was applied to Sephadex LH-20 column chromatography using MeOH followed by semi-preparative HPLC at a flow rate of 4 mL/min with a MeCN-H_2_O isocratic system (40:60, *v*/*v*) to obtain compound **15**. Subfraction M5 was applied to Sephadex LH-20 column chromatography with MeOH followed by semi-preparative HPLC with MeCN-H_2_O with a 0.1% formic acid isocratic system (30:70, *v*/*v*) to yield compound **12**.

#### 3.3.1. Corynol (**1**)

Pale brown amorphous powder, ${\left[\alpha \right]}_{D}^{20}$–9.7 (*c* 0.1, MeOH); IR (KBr) ν_max_ 3365, 1603, 1509, 1441, 1298, 1224 cm^−1^; UV (MeOH) λ_max_ (log ε) 230 (3.3), 280 (1.8) nm; ECD (MeOH) λ_max_ (Δε) 230 (−10.0), 280 (−0.7) nm; NMR data, see Table 1; HRESIMS *m*/*z* 283.1332 [M-H]^−^ (calcd for C_18_H_19_O_3_, 283.1334).

#### 3.3.2. 3′-Methoxycorynol (**2**)

Pale brown amorphous powder, ${\left[\alpha \right]}_{D}^{20}$–8.5 (*c* 0.1, MeOH); IR (KBr) ν_max_ 3399, 1603, 1514, 1446, 1367, 1229, 1208, 1151 cm^−1^; UV (MeOH) λ_max_ (log ε) 230 (sh), 280 (2.0) nm; ECD (MeOH) λ_max_ (Δε) 230 (−10.7), 280 (−1.5) nm; NMR data, see Table 1; HRESIMS *m*/*z* 313.1485 [M-H]^−^ (calcd for C_19_H_21_O_4_, 313.1440).

#### 3.3.3. 3′-Deoxycorynol (**3**)

Pale brown amorphous powder, ${\left[\alpha \right]}_{D}^{20}$–10.3 (*c* 0.1, MeOH); IR (KBr) ν_max_ 3355, 1613, 1509, 1436, 1343, 1229 cm^−1^; UV (MeOH) λ_max_ (log ε) 230 (3.0), 280 (1.5) nm; ECD (MeOH) λ_max_ (Δε) 230 (−8.0), nm; NMR data, see Table 1; HRESIMS *m*/*z* 267.1388 [M-H]^−^ (calcd for C_18_H_19_O_2_, 267.1385).

### 3.4. MTT Cell Viability Assay

Cell viability was assessed using an MTT (3-(4,5-dimethylthiazol-2-*yl*)-2,5-diphenyltetrazolium bromide)-based cytotoxicity assay. HEK293 cells were seeded in 10^4^ cells/well in 96-well plates and allowed to adhere for 24 h prior to compound treatment. Cells were treated at various concentrations in 96-well plates and incubated for 24 h. The final concentration of DMSO in the culture medium was maintained at 0.05% to avoid solvent toxicity. Subsequently, 20 μL of the 2 mg/mL MTT solution was added to each well of the plate and incubated for 2 h. Then, the absorbance was measured at 570 nm using a Versamax microplate reader (Molecular Devices Corporation, Sunnyvale, CA, USA). The percentage of cell viability is expressed as inversely proportional to the toxicities of the compounds, meaning that the higher the toxicity, the lower the cell viability. Cell viability is defined as the absorbance in the experimental well compared to that in the DMSO control wells.

### 3.5. Cell Culture and Transfection

HEK293 cells were kept in a 37 °C incubator with 5% CO_2_ and cultured in DMEM containing 10% fetal bovine serum and 1% penicillin/streptomycin. The cells were seeded in 24-well plates at a concentration of 10^4^ cells/well medium in each well. After 24 h, the cells were transfected with 0.2 μg pGL3-luc plasmid, 0.2 μg RSV-β-gal plasmid, 0.1 μg myc tagged p53 plasmid (myc-p53) and 0.2 μg flag-tagged SIRT1 plasmid (flag-SIRT1) using the PEI transfection reagent (Polyscience, Inc., Warrington, PA, USA). The cells were treated with this mixture for 5 h and then replaced with DMEM supplemented with 10% FBS for 24 h.

### 3.6. In Vitro SIRT1 Deacetylation in a Luciferase Reporter Assay

Transfected cells were treated with the compounds of interest 24 h after transfection. Using an analytical luminometer, luciferase activity was measured based on the addition of 30 μL of luciferin into 70 μL of lysate. A luciferase assay kit (Promega, Madison, WI, USA.) was used to check the promoter activity on the basis of measured luciferase. The β-galactosidase assay was then performed after adding the β-galactosidase substrate (Invitrogen) and incubated at 37 °C for 1 h. Substrate cleavage was quantified by measuring the optical density at 420 nm using a VersaMax microplate reader. Calculation of normalized values was performed by dividing luciferase activity by Renilla luciferase activity.

### 3.7. NAD^+^/NADH Ratio Measurement

The NAD*^+^*/NADH ratio was measured using the NAD*^+^*/NADH quantification kit (Biovision, Milpitas, CA, USA) from whole-cell extracts of HEK293 cells overexpressing SIRT1. Briefly, HEK293 cells were seeded at a density of 4 × 10^4^ cells per well in 12-well plates. Following 48 h of incubation, the cells were transfected with flag-tagged Sirt1 plasmid for 5 h and starved in DMEM containing 2% FBS overnight. Then, the cells were treated with vehicle, resveratrol or the compounds of interest for 12 h before analyzing the NAD^+^-to-NADH ratio according to the manufacturer’s instructions.

### 3.8. SIRT1 Deacetylation in a SIRT1 Enzyme-Based Assay

The SIRT1 deacetylation assay in a final reaction volume of 25 μL per well was conducted in half-volume 96-well microplates. Enzyme (0.5 U) with or without the test compound in Sirt1 assay buffer (1 mg/mL BSA, 1 mM MgCl_2_, 2.7 mM KCl, 137 mM NaCl and 50 mM Tris-HCl pH 8.0) was pre-incubated for 5 min at room temperature. The reaction was initiated by adding 500 μM NAD^+^ and 100 μM Fluor de Lys-SIRT1 (Biomol, Plymouth Meeting, PA, USA) and then incubated at room temperature for 1 h. Following incubation, the reaction was quenched with the addition of 25 μL of a solution containing 1X Fluor de Lys Developer II (Biomol, Plymouth Meeting, PA, USA) and 2 mM nicotinamide in HDAC assay buffer (1 mM MgCl_2_, 2.7 mM KCl, 137 mM NaCl and 50 mM Tris-HCl pH 8.0). After 30 min of incubation, 50 μL of the SIRT1 assay buffer was added to each well, the fluorophore generated in the reaction was excited with 360 nm light, and the emitted light (460 nm) was detected on a SpectraMax GEMINI XPS microplate reader (Molecular Devices, Sunnyvale, CA, USA).

### 3.9. Molecular Docking Simulation

All docking calculations were performed using MOE v2019.0102 software. Three different crystal structures of the human SIRT1 protein with PDB ID, 4KXQ, 4I5I and 5BTR were obtained from the RCSB Protein Data Bank (https://www.rcsb.org, 3 May 2020) and directly employed by MOE for docking calculations. Afterward, the protein structures were prepared and optimized by protonation and energy minimization using the LigX function in MOE with R-field solvation and Amber10: EHT as a forcefield. The docking ligands underwent preparation and energy minimization by MOE, with MMFF94x modified as a force field. The docking results were manipulated using the GBVI/WSA dG scoring function with the generalized Born solvation model (GBVI).

## 4. Conclusions

In our study to discover SIRT1 activation enhancers from *Corylopsis coreana*, three undescribed 8,3′-neolignans, namely corynol (**1**), 3-methoxycorynol (**2**) and 3′-deoxycorynol (**3**), together with 13 known compounds, were isolated from the methanolic extract of *C. coreana*. Among the isolated compounds, compound **3** showed a good stimulatory effect of SIRT1 regulation, and this is the first report for this activity for an 8,3′-neolignan. Ensuing docking studies suggested that compound **3** had potential binding affinity to several SIRT1 binding pockets. Thus, these results revealed 3′-deoxycorynol (**3**) as an SIRT1 activator.

## Figures and Tables

**Figure 1 plants-10-01684-f001:**
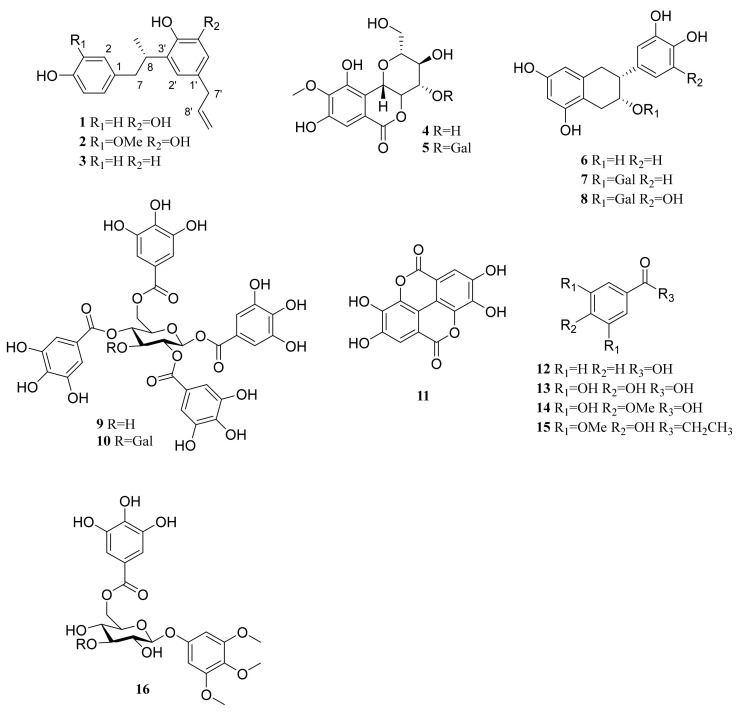
Chemical structures of 16 isolated compounds from *C. coreana*.

**Figure 2 plants-10-01684-f002:**
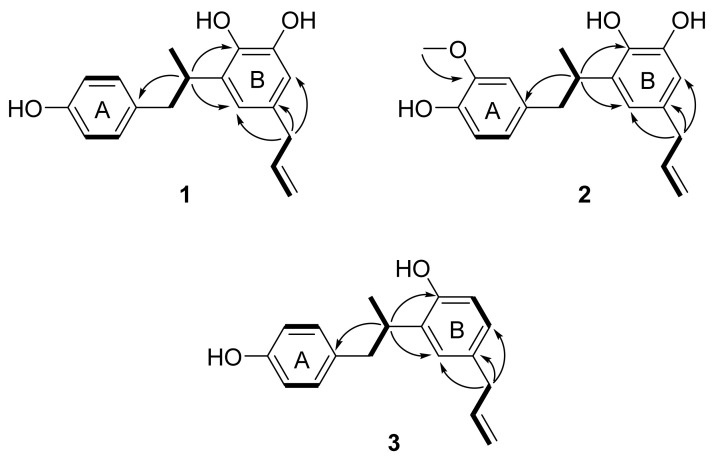
Key ^1^H-^1^H COSY (bold) and HMBC (arrows) correlations for compounds **1**–**3**.

**Figure 3 plants-10-01684-f003:**
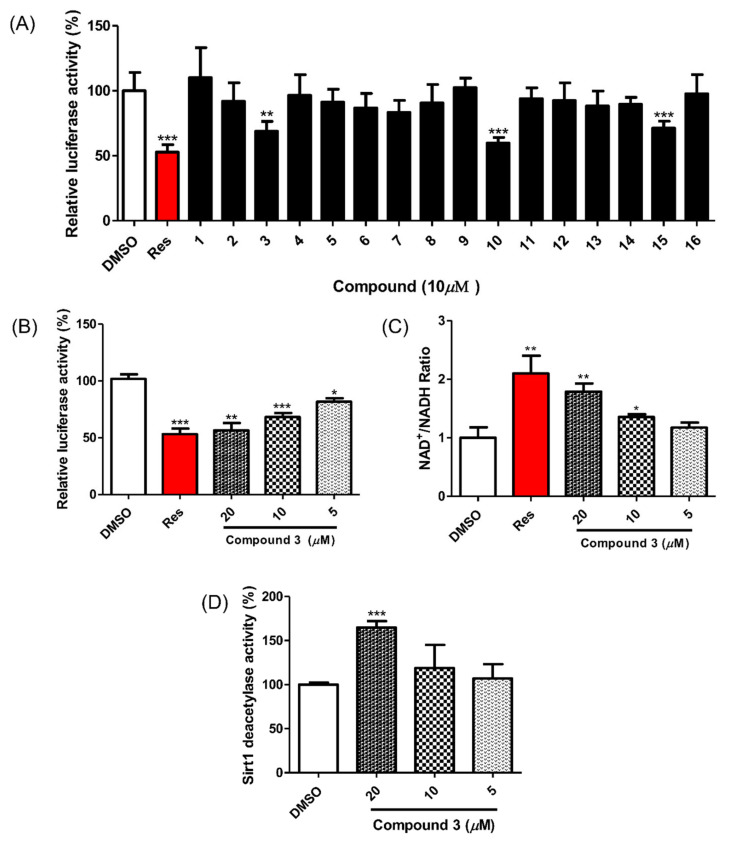
Effect of compounds **1**–**16** on SIRT1 deacetylase activity. (**A**) Compounds **3**, **10** and **16** reduced p53-mediated transcriptional activity. Test compounds (10 μM), vehicle or resveratrol (positive control) were applied for 24 h to HEK293 cells that were cotransfected with PG13-luc, myc-p53, flag-SIRT1 and RSV-β-gal plasmids. The fluorescent signal was detected via a firefly luciferase assay. (**B**) Compound **3** increased SIRT1 deacetylase activity in a dose-dependent manner in HEK293 cells. (**C**) Compound **3** promoted the intracellular NAD^+^/NADH ratio in HEK293 cells overexpressing SIRT1 after 12 h of treatment. (**D**) Compound **3** (20 μM) enhanced SIRT1 deacetylase activity in the enzyme-based assay. Test compounds were preincubated with SIRT1 enzyme and then reacted with Fluor de Lys-SIRT1 deacetylase substrate for 1 h. Data are presented as the mean ± SD (*n* = 3), * *p* < 0.05, ** *p* < 0.01 and *** *p* < 0.001 compared to the vehicle-treated samples.

**Figure 4 plants-10-01684-f004:**
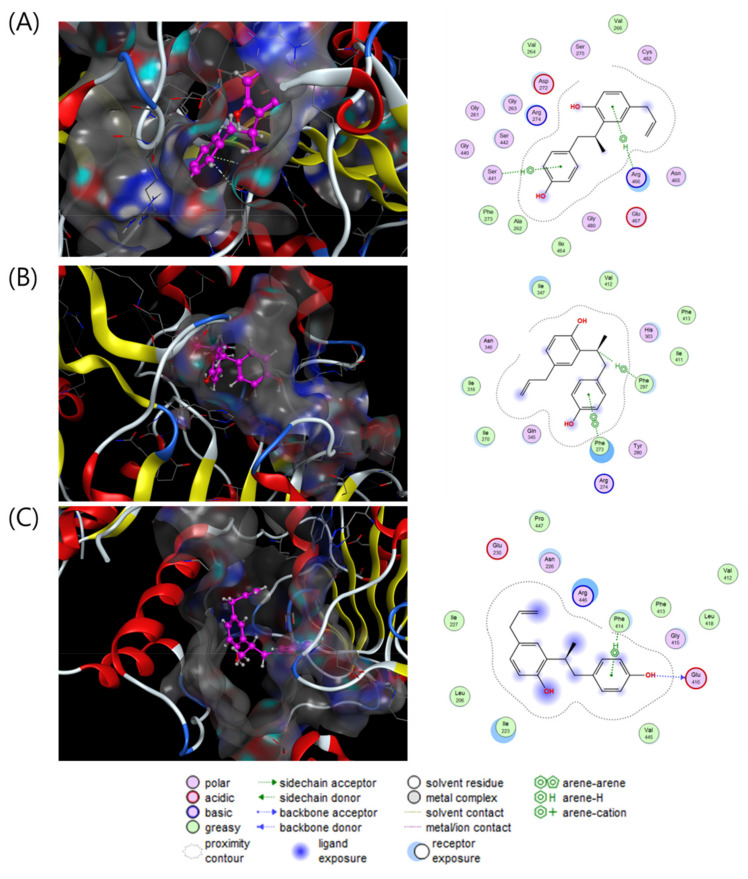
Interaction analysis of compound **3** in the binding pockets of 4KXQ (**A**), 4I5I (**B**) and 5BTR (**C**). Docking poses of compound **3** (violet purple) in the binding pockets (gray).

**Table 1 plants-10-01684-t001:** ^1^H and ^13^C NMR spectroscopic data of compounds **1**–**3**.

	1		2		3	
Position	δ_C_, Type	δ_H_ (*J* in Hz)	δ_C_, Type	δ_H_ (*J* in Hz)	δ_C_, Type	δ_H_ (*J* in Hz)
1	133.7, C	-	134.3, C	-	133.6, C	-
2	131.1, CH	6.95, d (8.4)	113.8, CH	6.61, brs	131.1, CH	6.93, d (7.9)
3	115.7, CH	6.64, d (8.4)	148.4, C	-	115.7, CH	6.63, d (7.9)
4	156.2, C	-	145.3, C	-	156.2, C	-
5	115.7, CH	6.64, d (8.4)	115.6, CH	6.63, dd (8.0, 1.2)	115.7, CH	6.63, d (7.9)
6	131.1, CH	6.95, d (8.4)	122.7, CH	6.57, d (8.0)	131.1, CH	6.93, d (7.9)
7	43.4, CH_2_	2.88, dd (13.5, 5.8) 2.55, dd (13.5, 8.8)	43.8, CH_2_	2.89, dd (13.4, 6.4) 2.60, dd (13.4, 8.1)	43.4, CH_2_	2.88, dd (13.5, 5.8) 2.55, dd (13.5, 8.7)
8	35.9, CH	3.33, m	35.9, C	3.35, m	35.8, C	3.33, m
9	19.9, CH_3_	1.11, d (6.9)	20.2, CH_3_	1.14, d (6.4)	19.8, CH_3_	1.12, d (7.0)
1′	132, C	-	132.0, C	-	131.8, C	-
2′	119, CH	6.44, d (1.9)	119.2, CH	6.43, brs	127.5, CH	6.78, dd (8.3, 2.0)
3′	145.8, C	-	134.6, C	-	134.4, C	-
4′	141.9, C	-	142.0, C	-	153.9, C	-
5′	134.8, C	-	145.8, C	-	115.9, CH	6.65, d (8.3)
6′	113.7, CH	6.47, d (1.9)	113.7, CH	6.46, brs	128.4, CH	6.88, d (2.0)
7′	40.9, CH_2_	3.19, d (6.7)	40.9, CH_2_	3.19, d (6.8)	40.7, CH_2_	3.24, d (6.5)
8′	139.7, CH	5.90, ddt (16.8, 10.2, 6.7)	139.7, CH	5.90, m	139.9, CH	5.91, m
9′	115.1, CH_2_	5.00, dd (16.8, 1.1)	115.1, CH_2_	5.00, brd (17.0)	115.1, CH_2_	4.98, brd (18.0)
		4.97, dd (10.2, 1.1)		4.97, brd (10.2)		4.97, brd (8.5)
3-OCH_3_	-	-	56.2, CH_3_	3.73, s	-	-

## Data Availability

Data is contained within the article or Appendix A.

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
