# Peer review of "SIRT1 Activation Enhancing 8,3′-Neolignans from the Twigs of Corylopsis coreana Uyeki"

_plants, 2021, doi:10.3390/plants10081684_

Round 1
Reviewer 1 Report
The manuscript describes the phytochemical study of the twigs of
Korean corylopsis Uyeki as well as SIRT1 activation enhancing by three new neolignans found in study.
In the introduction, the authors do not mention that similar compounds have been isolated from Myristica fragrans in the following references.
Helvetica Chimica Acta (2007), 90(8), 1491-1496
Biomed. Chromatogr. 2012; 26: 703–707
Journal of Pharmaceutical and Biomedical Analysis 145 (2017) 725–733
The authors should check the following list of references to complete the previous studies carried out on the properties of the plant studied:
HPLC analysis, optimization of extraction conditions and biological evaluation of Corylopsis coreana uyeki flos
By: Seo, Ji-Hye; Kim, Jung-Eun; Shim, Jung-Hyun; Yoon, Goo; Bang, Mi-Ae; Bae, Chun-Sik; Lee, Kyung-Jin; Park, Dae-Hun; Cho, Seung-Sik
From Source: Molecules (2016), 21(1), 94/1-94/13
Anti-oxidative and anti-proliferative activity on human prostate cancer cells lines of the phenolic compounds from Corylopsis coreana Uyeki
By: Kim, Manh Heun; Ha, Sung Yi; Oh, Myeong Hwan; Kim, Han Hyuk; Kim, So Ra; Lee, Min Won
From Source: Molecules (2013), 18, 4876-4886
Effects of extracts from Corylopsis coreana Uyeki (Hamamelidaceae) flos on xanthine oxidase activity and hyperuricemia
By: Yoon, In-Soo; Park, Dae-Hun; Ki, Sung-Hwan; Cho, Seung-Sik
From Source: Journal of Pharmacy and Pharmacology (2016), 68(12), 1597-1603
The chalcone compound isosalipurposide (ISPP) exerts a cytoprotective effect against oxidative injury via Nrf2 activation
By: Han, Jae Yun; Cho, Seung Sik; Yang, Ji Hye; Kim, Kyu Min; Jang, Chang Ho; Park, Da Eon; Bang, Joon Seok; Jung, Young Suk; Ki, Sung Hwan
From Source: Toxicology and Applied Pharmacology (2015), 287(1), 77-85
Chalcone glycoside in the flowers of six Corylopsis species as yellow pigment
By: Iwashina, Tsukasa; Takemura, Tomoko; Mishio, Tamaki
From Source: Journal of the Japanese Society for Horticultural Science (2009), 78(4), 485-490
-------------------------------
Flavonoids from the leaves of six Corylopsis species (Hamamelidaceae)
By: Iwashina, Tsukasa; Kitajima, Junichi; Takemura, Tomoko
From Source: Biochemical Systematics and Ecology (2012), 44, 361-363
New application of extract of Corylopsis Coreana (Uyeki) for preparing skin care products
By: Lee, Gang Tae; Lee, Jeong No; Lee, Gwang Sik; Lee, Seung Ji; Lee, Geon Guk
From Source: Korea, Republic of, KR2009002543 A 2009-01-09
Characterization of the Corylopsis coreana using molecular markers
By: Roh, Mark S.; Lee, Ae Kyung; Choi, Ik Young; Kim, Jae Yeong; Joung, Young He; Lee, Sun Ha; Suh, Jeung Keun
From Source: Horticulture, Environment and Biotechnology (2007), 48(3), 176-187
Anti-inflammatory activity of Corylopsis coreana uyeki extracts and isolated compounds by inhibiting nitric oxide and cytokine production
By: Yin, Jun; Wang, Hyesoo; Lee, Minwon
From Source: Pharmacognosy Magazine (2019), 15(61), 317-321
Health food composition comprising ginseng, Salicornia europaea, Corylopsis coreana, and Erythronium japonicum
By: Jung, Mi Ran; Choi, Gyeong Min; Choi, Mi Rae; Cha, Jeong Dan
From Source: Korea, Republic of, KR2015016449 A 2015-02-12
Extracts from Erythronium japonicum and Corylopsis coreana Uyeki reduce 1,3-dichloro-2-propanol-mediated oxidative stress in human hepatic cells
By: Bae, Chun-Sik; Yun, Chul-Ho; Ahn, Taeho
From Source: Food Science and Biotechnology (2019), 28(1), 175-180
Antibacterial composition comprising extract of Corylopsis coreana
By: Cho, Seung Sik; Bang, Mi Ae; Bae, Min Seok
From Source: Korea, Republic of, KR2015137929 A 2015-12-09
Anti-inflammatory compositions comprising Corylopsis coreana leaf extract or phenolic compounds derived from the extract
By: Lee, Min Won; Kim, Manh Heun
From Source: Korea, Republic of, KR2013086448 A 2013-08-02
Anti-inflammatory compositions containing extracts of Corylopsis coreana and Erythronium japonicum
By: Jung, Ui Su; Yoon, Jae Ung; Park, Min Hui; Moon, Je Bong
From Source: Korea, Republic of, KR1382137 B1 2014-04-07
--------------------------------
Identification and extraction optimization of active constituents in Citrus junos Seib ex TANAKA peel and its biological evaluation
By: Shim, Jung-hyun; Chae, Jung-il; Cho, Seung-sik
From Source: Molecules (2019), 24(4), 680
Hepatoprotective compositions containing a compound isolated from Corylopsis coreana extracts
By: Ki, Seong Hwan; Bang, Mi Ae; Cho, Seung Sik
From Source: Korea, Republic of, KR2016034133 A 2016-03-29
Antimicrobial composition comprising natural plant extract as effective ingredient
By: Song, Won Seop; Bu, Hui Ok; Kim, Dong Hun
From Source: Korea, Republic of, KR2014018589 A 2014-02-13
Food composition having anti-obesity, anti-cancer, anti-oxidative and immune function enhancing activities and utilized for producing granule tea and roasted green tea and increasing activity of B cells and T cells
By: Bu, Hui Ok; Jung, Ui Su; Choi, Gyeong Min; Bang, Mi Ae; Song, Won Seop
From Source: Korea, Republic of, KR2015007037 A 2015-01-20
Anti-obesity and anti-diabetic health food composition with excellent glucose-lowering effect
By: Bu, Hui Ok; Kim, Hak Hyeon; Shin, Ji San
From Source: Korea, Republic of, KR2014009717 A 2014-01-23
Health food composition with antioxidant activity
By: Bu, Hui Ok; Kim, Hak Hyeon; Jung, Ui Su; Choi, Gyeong Min; Bang, Mi Ae; Song, Won Seop
From Source: Korea, Republic of, KR2015127849 A 2015-11-18
Antimicrobial and anti-biofilm activities of the methanol extracts of medicinal plants against dental pathogens Streptococcus mutans and Candida albicans
By: Choi, Hyoung-An; Cheong, Dae-Eun; Lid, Ho-Dong; Kim, Won-Ho; Had, Mi-Hyoun; Oh, Myung-Hwan; Wu, Yuanzheng; Shin, Hyun-Jae; Kim, Geun-Joong
From Source: Journal of Microbiology and Biotechnology (2017), 27(7), 1242-1248
----------------------------------
Screening of antioxidative activities and antiinflammatory activities in local native plants
By: Kim, Han Hyuk; Kwon, Joo Hee; Park, Kwan Hee; Kim, Manh Heun; Oh, Myoeng Hwan; Choe, Kang In; Park, Sang Hee; Jin, Hye Young; Kim, Sung Sik; Lee, Min Won
From Source: Saengyak Hakhoechi (2012), 43(1), 85-93
--------------------------
Sophora subprostrata extracts for inhibiting activity of acetylcholinesterase
By: Ju, Han Seung; Won, Mu Ho; Yoo, Gi Yeon; Li, Hwa; Kim, Hyeon Seop; Kim, Sang Beom; Choi, Jang Won
From Source: Korea, Republic of, KR2011082444 A 2011-07-19
------------------
Antimicrobial and Anti-Inflammatory Effects of Ethanol Extract of Corylopsis coreana Uyeki Flos.
By: Park, Da-Eon; Yoon, In-Soo; Kim, Jung-Eun; Seo, Ji-Hye; Yoo, Jin-Cheol; Bae, Chun-Sik; Lee, Chang-Dai; Park, Dae-Hun; Cho, Seung-Sik
From Source: Pharmacognosy magazine (2017), 13(50), 286-292
Composition containing corylopsis coreana hairy extract for preventing hair loss
By: Kwon, Hyeok Cheol; Kim, Tae Yang; Lee, Gang Tae; Lee, Geon Guk
From Source: Korea, Republic of, KR1793493 B1 2017-11-06
Protective effects of Erythronium japonicum and Corylopsis coreana Uyeki extracts against 1,3-dichloro-2-propanol-induced hepatotoxicity in rats.
By: Kim, Seunghyun; Boo, Hee-Ock; Ahn, Taeho; Bae, Chun-Sik
From Source: Applied microscopy (2020), 50(1), 29
Optimized Extract from Corylopsis coreana Uyeki (Hamamelidaceae) Flos Inhibits Osteoclast Differentiation.
By: Lee, Yongjin; Kim, Jung-Eun; Kim, Kwang-Jin; Cho, Seung-Sik; Son, Young-Jin
From Source: Evidence-based complementary and alternative medicine : eCAM (2018), 2018, 6302748
Extract of corylopsis coreana and tellimagrandin i isolated therefrom having antifungal activity by inhibiting enzymes related to glyoxylate cycle without side effects
By: Kim, Sung Uk; Kang, Tae Hoon; Park, Ki Duk; Kim, Sung Eun; Choi, Won Shik; Lee, Yun Mi; Moon, Jae Sun; Park, Ho Yong
From Source: Korea, Republic of, KR729437 B1 2007-06-27
The complete chloroplast genome sequence of Corylopsis multiflora Hance var. nivea Chang.
By: Lv, Ting; Chen, Shuifei; Zhao, Rong; Wang, Ningjie; Fang, Yanming
From Source: Mitochondrial DNA. Part B, Resources (2021
, 6(1), 271-273
Author Response
Reviewer 1.
The manuscript describes the phytochemical study of the twigs of Korean corylopsis Uyeki as well as SIRT1 activation enhancing by three new neolignans found in study.
In the introduction, the authors do not mention that similar compounds have been isolated from Myristica fragrans in the following references.
Helvetica Chimica Acta (2007), 90(8), 1491-1496
Biomed. Chromatogr. 2012; 26: 703–707
Journal of Pharmaceutical and Biomedical Analysis 145 (2017) 725–733
> Thank you for your kind suggestion. We cited two papers (Journal of Pharmaceutical and Biomedical Analysis 145 (2017) 725–733, Helvetica Chimica Acta (2007), 90(8), 1491-1496) in the introduction and discussion sections as: “and also found in the metabolism of dehydrodiisoeugenol (DDIE) with demethylation and ring-opening reaction [16].” and “1-Deoxycarinatone, which has the same 8,3′-neolignan structure as that of 1 showed a negative Cotton effect at 285 nm in its ECD spectrum [31] [32].”
The authors should check the following list of references to complete the previous studies carried out on the properties of the plant studied.
> Thank you for your kind suggestion. We checked your suggested list of references.

Reviewer 2 Report
The manuscript described the isolation, structural determination of three undescribed neolignans from Corylopsis coreana, a member of Hamamelidaceae family. Additionally, the stimulation effects on SIRT1, which is associated to aging delay process.
The structural determination of the compounds was made properly based on spectroscopic means. The only minor mistakes are the coupling constants that are not the same. For instance, compound 1, H7: 13.4 versus 13.5; 8.8 versus 8.4, H2’ and H6’ (1.9 versus 2.1); compound 3, H7: 13.5 versus 13.4, and 7.9 versus 8.7 (?); 8.3 versus 8.1, 2.2 versus 2.1. The differences were due to the low digital points, but the authors should adopt the same value for the coupled system.
I have a few minor revisions as follows:
Line 233, provide the mass of aqueous residues submitted to the HP-20 resin column chromatography.
Lines 274 and 279, for consistency, capitalize the first letter of compounds name. Check for typos.
Author Response
Reviewer 2.
The manuscript described the isolation, structural determination of three undescribed neolignans from Corylopsis coreana, a member of Hamamelidaceae family. Additionally, the stimulation effects on SIRT1, which is associated to aging delay process.
The structural determination of the compounds was made properly based on spectroscopic means. The only minor mistakes are the coupling constants that are not the same. For instance, compound 1, H7: 13.4 versus 13.5; 8.8 versus 8.4, H2’ and H6’ (1.9 versus 2.1); compound 3, H7: 13.5 versus 13.4, and 7.9 versus 8.7 (?); 8.3 versus 8.1, 2.2 versus 2.1. The differences were due to the low digital points, but the authors should adopt the same value for the coupled system.
> Thank you for your kind advice. We checked them and fixed them without 8.8 versus 8.4 on compound 1, 7.9 versus 8.7 on compound 3. The coupling constant of 8.4 is from the ortho coupling between H-2,6 and H-3,5. So it is not correlated with H-7 which is located on the outside of aromatic system. In the same manner, the coupling constant of 7.9 is from the ortho coupling between H-2,6 and H-3,5.
I have a few minor revisions as follows:
Line 233, provide the mass of aqueous residues submitted to the HP-20 resin column chromatography.
> We added the amount of aqueous residues in the manuscript as: “The aqueous residues were subjected to HP-20 resin column chromatography eluting with water (49 g) and methanol (12.2g) to provide two fractions.”
Lines 274 and 279, for consistency, capitalize the first letter of compounds name. Check for typos.
> Thank you for your kind suggestions. We thoroughly rechecked the manuscript for typos.

Reviewer 3 Report
The manuscript describes isolation, structural characterization and bioactivity testing of several 8,3'-neolignans isolated from Corylopsis coreana. Three of them have not been described yet.
It is a well done study, however without much really new. All procedures used in this study were carried out very carefully and are well documented. The manuscript has been written in very good English without typographical and grammar errors. I have managed to find only one (l. 156, text - tested). There are no conclusions, which are not supported by experimental results.
I would recommend extending the study by docking molecule 1 and 2 and look if their lower biological activity is reflected in worsening the binding parameters. Despite my objections, I recommend the manuscript for publication.
Author Response
Reviewer 3.
The manuscript describes isolation, structural characterization and bioactivity testing of several 8,3’-neolignans isolated from Corylopsis coreana. Three of them have not been described yet.
It is a well done study, however without much really new. All procedures used in this study were carried out very carefully and are well documented. The manuscript has been written in very good English without typographical and grammar errors. I have managed to find only one (l. 156, text – tested). There are no conclusions, which are not supported by experimental results.
> Thank you for your advice. Typo error pointed by reviewer was corrected. We described our conclusion on section 4, but we wrote an additional sentence as: “ Thus, these results strongly indicate that 3′-deoxycorynol (3) is a new SIRT1 activator.”
I would recommend extending the study by docking molecule 1 and 2 and look if their lower biological activity is reflected in worsening the binding parameters. Despite my objections, I recommend the manuscript for publication.
> Thank you for your kind review and clear decision. Recently, we didn’t test the docking analysis with compounds 1 and 2, but it is interesting point. In our further research, we will consider that point.
